# A Comprehensive Review on the Role of Resting-State Functional Magnetic Resonance Imaging in Predicting Post-Stroke Motor and Sensory Outcomes

Foteini Christidi [1,†], Ilias Orgianelis [1,†], Ermis Merkouris [1], Christos Koutsokostas [1], Dimitrios Tsiptsios [1,*], Efstratios Karavasilis [2], Evlampia A. Psatha [2], Anna Tsiakiri [1], Aspasia Serdari [3], Nikolaos Aggelousis [4] and Konstantinos Vadikolias [1]

1   Neurology Department, School of Medicine, Democritus University of Thrace, 68100 Alexandroupolis, Greece; christidi.f.a@gmail.com (F.C.); iliaorgi@med.duth.gr (I.O.); ermimerk@med.duth.gr (E.M.); chriskoutsokostas2001@gmail.com (C.K.); anniw_3@hotmail.com (A.T.); vadikosm@yahoo.com (K.V.)
2   Department of Radiology, School of Medicine, Democritus University of Thrace, 68100 Alexandroupolis, Greece; ekaravas@med.duth.gr (E.K.); eviepsatha@yahoo.gr (E.A.P.)
3   Department of Child and Adolescent Psychiatry, School of Medicine, Democritus University of Thrace, 68100 Alexandroupolis, Greece; aserntar@med.duth.gr
4   Department of Physical Education and Sport Science, Democritus University of Thrace, 69100 Komotini, Greece; nagelous@phyed.duth.gr
*   Correspondence: tsiptsios.dimitrios@yahoo.gr
†   These authors contributed equally to this work.

**Abstract:** Stroke is a major leading cause of chronic disability, often affecting patients' motor and sensory functions. Functional magnetic resonance imaging (fMRI) is the most commonly used method of functional neuroimaging, and it allows for the non-invasive study of brain activity. The time-dependent coactivation of different brain regions at rest is described as resting-state activation. As a non-invasive task-independent functional neuroimaging approach, resting-state fMRI (rs-fMRI) may provide therapeutically useful information on both the focal vascular lesion and the connectivity-based reorganization and subsequent functional recovery in stroke patients. Considering the role of a prompt and accurate prognosis in stroke survivors along with the potential of rs-fMRI in identifying patterns of neuroplasticity in different post-stroke phases, this review provides a comprehensive overview of the latest literature regarding the role of rs-fMRI in stroke prognosis in terms of motor and sensory outcomes. Our comprehensive review suggests that with the advancement of MRI acquisition and data analysis methods, rs-fMRI emerges as a promising tool to study the motor and sensory outcomes in stroke patients and evaluate the effects of different interventions.

**Keywords:** functional magnetic resonance imaging; resting-state functional connectivity; stroke; motor outcomes; sensory outcomes; biomarkers

## 1. Introduction

Stroke is one of the leading causes of chronic disability due to its impact on the motor, sensory, and cognitive functions of patients. The presence of these deficits is one of the factors that hinder the full recovery of patients [1]. During the post-stroke period, the brain inherently tries to restore its functional status that has been affected [2]. There can be recovery, where the affected function is fully restored, or compensation, where mechanisms emerge through which other brain regions participate in achieving a previous level of functionality [3]. However, compensatory mechanisms may sometimes lead to incomplete recovery, exacerbating the existing deficit through maladaptive plasticity [4]. The range of these mechanisms partially explains the range of varying degrees of effectiveness of intervention programs [5]. It also highlights the need to study additional factors or markers

that could contribute to designing better rehabilitation programs, predicting motor and sensory outcomes, and better assessing intervention effectiveness.

Functional magnetic resonance imaging (fMRI) is the most commonly used method of functional neuroimaging, allowing the non-invasive study of brain activity [6]. It is based on the investigation of the differences between the paramagnetic properties of oxygenated and non-oxygenated hemoglobin, examined through the blood oxygen level dependent (BOLD) signal in different brain regions either during a task (task fMRI) or at rest (resting state fMRI; rs-fMRI) within the MRI scanner. During rs-fMRI, the subject remains still and calm inside the MRI scanner with eyes usually closed, without receiving external stimuli or thinking about anything. The time-dependent coactivation of different brain regions at rest is described as resting-state (rs) activation. Distinct brain areas that are co-activated in response to stimuli (e.g., during task-fMRI) also show specific or functional organization at rest and constitute the so-called rs functional networks (Figure 1). Thus, rs-fMRI is a task-independent approach that is based on intrinsic low-frequency fluctuations (typically <0.1 Hz) in the BOLD signal. The latter can be used to compute the functional connectivity between spatially remote areas based on the temporal correlations of the BOLD signals between these areas [7]. These functionally connected networks are named after the function they are related to when activated [8]. Each rs functional network is composed of individual brain regions (structures/nodes), whose functional connectivity (FC) is based on the quantitative correlation of the perfusion rate of individual structures. The magnitude of the correlations of the BOLD signal time series between nodes constituting a functional structure and between individual structures constituting a network is considered an indication of the adequacy of FC [9]. Two of the main ways of detecting FC between individual structures/nodes are: (a) correlation of the BOLD signal time series of individual structures, (b) independent component analysis. As a non-invasive task-independent functional neuroimaging approach, rs-fMRI may therefore provide therapeutically useful information on both the focal vascular lesion and the connectivity-based reorganization and subsequent functional recovery in stroke patients [10].

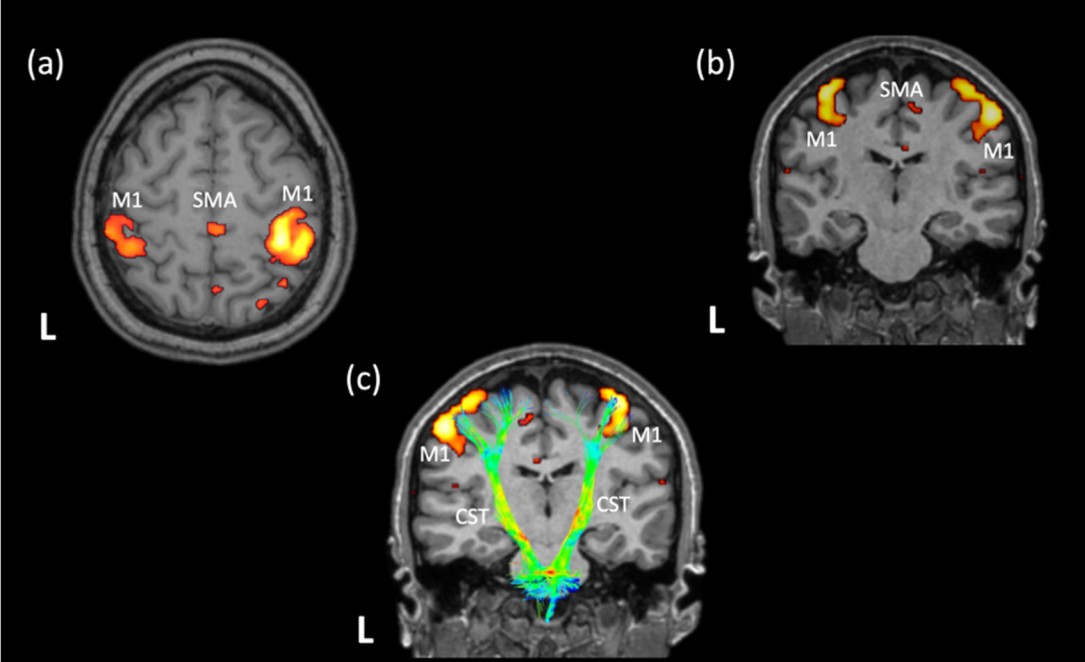

**Figure 1.** The characteristic patterns of functional connectivity at rest between the regions of the motor network of a healthy individual. The nodes of the motor network (left and right M1, SMA) are overlaid on a 3D-T1 sequence and presented in the axial (**a**) and coronal (**b**) planes. The locations of the left and right M1 nodes and their association with the distribution of the major motor white-matter pathway (CST) is presented in (**c**). Note that the CST had been reconstructed using tractography on

available DTI data for the same subject and is colored based on the distribution of the fractional anisotropy values along the tract. The analysis of the DTI data, as well as the rs-fMRI data and the identification of the motor network at rest, have been performed on a healthy subject (who provided written informed consent for the data acquisition, analysis, and presentation) using the Brainance MD platform (Advantis Medical Imaging). rs-fMRI = resting-state functional magnetic resonance imaging; DTI = diffusion tensor imaging; L = left hemisphere; M1 = primary motor cortex; SMA = supplementary motor area; CST = corticospinal tract.

Considering the role of prompt and accurate prognosis in stroke survivors along with the potential of rs-fMRI in identifying patterns of neuroplasticity in different post-stroke phases, this review provides a comprehensive overview of the latest literature regarding the role of rs-fMRI in stroke prognosis in terms of motor and sensory outcomes.

## 2. State-of-the-Art of Current Literature on rs-fMRI in Motor and Sensory Outcomes after Stroke

To address the main aim of this comprehensive review, we reviewed most representative human studies (cross-sectional or longitudinal) during the last decade which used rs-fMRI and focused on stroke motor and sensory outcomes (Supplementary Material: Supplementary Tables S1 and S2, Supplementary Figure S1).

### *2.1. Prediction of Motor Outcomes Using rs-fMRI*
2.1.1. Changes in FC Based on Traditional and Relatively New rs-fMRI Indices

The topological characteristics of the cortical motor-related network in patients with subcortical stroke were examined in a study by Yin et al. (2013) in which all areas of interest were distal to the initial lesion [11]. The results indicated decreased FC between the ipsilesional M1 and the contralesional middle frontal gyrus, as well as between the ipsilesional postcentral gyrus and the contralesional postcentral gyrus. Accordingly, disruption of interhemispheric FC in the somatomotor network was found to be strongly associated with post-stroke upper limb dysfunction, whereas it was not associated with performance in the same network. Additionally, the study identified diverse patterns of functional remodeling in both cortical connectivity and localized homogeneity, suggesting varying results in hand functionality. Specifically, decreased betweenness centrality (that assesses the amount of influence a node has over the flow of information in a graph) of the ipsilesional dorsolateral premotor cortex indicated poor outcomes in hand functionality.

According to Li et al. (2014), patients with dysphagia showed greater rs functional architecture irregularities than patients who experienced no difficulty swallowing [12]. These irregularities are largely related to alterations in the coherent intrinsic neuronal activity of BOLD fluctuations detected using rs-fMRI. The results imply that effective recovery is linked to brain activation that is relevant to cortical swallowing representation in the compensatory hemisphere or other recruited regions of the unaffected hemisphere.

Lam et al. (2018), with respect to FC as a stroke biomarker, investigated left (L) and right (R) motor cortex rs-FC (L M1-R M1), which they considered to be a sign of functional integrity [13]. Stroke patients with higher L M1-R M1 rs-connectivity were less disabled than those with lower connectivity, consistent with the fact that L M1-R M1 patterns at rest are comparable to those that arise during motor task performance and contribute to the variability of motor behaviors. Contrary to previous studies on this topic, in which both a positive and a negative relationship between the two were found, possibly because of different stroke parameters, there was no significant interaction between corticospinal tract (CST) injury and L M1-R M1 rs-connectivity in the regression models obtained in this study. According to the authors, this underscores the need for larger-scale studies that assess stroke patients at all stages of chronicity and severity. The authors speculate that CST injury and L M1-R M1 rs connectivity may serve as different indices of motor outcomes considering the above.

It is well documented that after rehabilitation, ipsilesional intrahemispheric FC decreases and interhemispheric FC increases. Furthermore, the decrease in FC between bilateral motor cortices correspond more strongly with neurological function than the decrease in FC inside ipsilesional motor cortices. Chi et al. (2018) studied patients with mild acute ischemic stroke (AIS) stroke and found that although the interhemispheric FC of the motor network was lower in patients than in healthy subjects, there were differences between patients with favorable and poor outcomes only in the M1 and in the contralesional dorsal premotor area [14]. The authors concluded that in functional recovery after AIS, interhemispheric FC may be more important than ipsilesional intrahemispheric FC. Additionally, FC may represent a potentially therapeutic target in stroke patients as well as a reliable and prompt imaging biomarker for AIS using neurostimulation techniques.

Zou et al. (2018) revealed that the frontoparietal system is essential for higher-order motor–cognitive processing, including motor imagery and the regulation of goal-directed movements, whereas the sensorimotor system is primarily active during motor execution and somatosensory information processing. Both systems have been found to be involved in cerebral remodeling, characterized by aberrant task-evoked brain activation as well as disturbed rs-FC [15]. The random reorganization hypothesis is supported by the fact that when a stroke affects the brain, connectivity reorganization occurs in two distinct phases: first, existing links are broken and rewired, and second, new links are created due to connections being established randomly across all possible brain regions. The authors also concluded that there was a correlation between the degree of motor deficits and lower interhemispheric FC between cortical motor regions, even in individuals who had made an excellent clinical recovery.

In the study conducted by Hong et al. (2019), only the completely paralyzed patients showed reduced FC in perceptual areas, although both patients with completely and partially paralyzed hands showed reduced within-network FC in the contralesional superior parietal cortex and ipsilesional supplementary motor area (SMA) [16]. Because continuous dominant stimulation of the contralesional sensorimotor cortex has been shown to interfere with normal function of the paretic hand, the authors suggested that this region may be involved in functional outcomes of the ipsilateral hand. They discovered that patients whose hands were completely paralyzed had increased network FC in the contralesional sensorimotor cortex compared with patients who were only partially paralyzed, suggesting that the severe hand impairments were followed by excessive mobilization of contralesional sensorimotor assets. In addition, there was a negative correlation between the Fugl–Meyer Assessment (FMA) scores and the mean FC in the contralesional sensorimotor cortex, indicating that contralesional hemisphere involvement is not a desirable indicator in stroke patients in the chronic stage.

More sophisticated approaches to rs-fMRI data have also been applied during the past few years. Liu et al. (2015) revealed that the amplitude of low-frequency fluctuation (ALFF), which reflects regional spontaneous neuronal activity, was increased in the bilateral M1 area during the initial phase of subcortical infarction [17]. The authors hypothesized that this is due to several factors, including altered vasomotor activity and neurovascular coupling, although the exact neurophysiological basis remains unknown. The findings indicate that increased spontaneous neuronal activity, determined via rs-fMRI, might possibly contribute to the functional reorganization of physically impaired brain regions during post-stroke motor recovery.

Moreover, in a study by Zhao et al. (2018), patients with completely paralyzed hands showed decreased regional homogeneity (ReHo) in the bilateral cerebellar posterior lobes and increased ReHo in the contralesional SMA compared with patients with less severe paralysis in the typical frequency band [18]. Similar findings were noted in the slow and subfrequency bands, supporting the authors' theory that the frequency of low-frequency oscillations and stroke severity influence spontaneous brain activity after stroke. In addition, there was a strong association between the mean ReHo values in these regions and the

FMA-HW scores in all patients, suggesting that frequency-specific changes in ReHo may be related to hand function recovery in stroke patients.

Brain entropy (BEN), an fMRI-based approach, was utilized by Liang et al. (2020) to map the temporal complexity of the entire brain and its capacity to handle incoming or outgoing data [19]. Lower BEN values often signify decreased neural activity and erratic information processing capabilities. It has been noted that lower BEN values in the contralesional precentral gyrus and ipsilesional M1 may be indicative of a reduced aptitude of information exploration in the contralesional sensorimotor system. Furthermore, lower BEN values were observed in the bilateral SMA, suggesting that the operational framework of motor execution and planning was disturbed. It is possible that decreased brain complexity in this area is connected to the impaired integrity of the cortical structure. Additionally, the BEN values in the ipsilesional SMA showed a positive correlation with the FMA scores, suggesting that the absence of brain complexity in this region could contribute to poor motor outcomes in stroke patients. This occurs because the SMA is essential for the progression of motor rehabilitation.

Additionally, Bonkhoff et al. (2020), in a study of patients with mild to severe stroke, demonstrated for the first time that moderately impaired patients spent significantly more time in state 2, a less connected state, while patients with severe motor symptoms significantly preferred state 1, a state with strong intradominal connectivity [20]. Unexpectedly, they discovered that moderately impaired patients had more pronounced changes in FC (between the ipsi- and contralesional sensorimotor cortices and the same sensorimotor network and bilateral paracentral cortex) in both static and dynamic analyses, despite milder clinical deficits. However, the researchers found impaired interhemispheric connectivity between sensorimotor cortical components in severe strokes, with motor symptoms only in dynamic analysis. Bonkhoff et al. (2021) showed that in the first three months after stroke, the observed recovery in stroke severity was strongly connected with—or showed strong trends in—dynamic connectivity between three pairs of networks [21]. The change in stroke severity and the dynamic connectivity of default mode networks in the bilateral intraparietal lobule and left angular gyrus were significantly inversely correlated in state 1. Dynamic connectivity of the bilateral putamen and superior temporal gyrus and first and bilateral anterior insula also showed positive associations with recovery of stroke severity at stage 3. Recovery of motor symptoms, which make up a large proportion of the NIHSS score, may be related to the subcortical putamen, revealing its importance in stroke outcomes. It is known that the default mode and cognitive control networks control basic brain activities, which explains why individuals who have lesions in these regions tend to have worse stroke outcomes. In addition, the strong negative connectivity between networks suggests that there is less communication between different functional areas. However, the researchers found no significant association between the degree of segregation and changes in stroke severity in the first three months after stroke.

2.1.2. Changes in FC and Their Association with the Application of Specific Rehabilitation Methods

In a study by Fan et al. (2015), who used robot-assisted bilateral arm therapy (RBAT) for stroke patients, changes in M1–M1 connectivity before and after treatment were associated with better motor and functional progress, and these differences were thought to serve as a significant mediator of the relationship between changes in some disability scores [22]. As a result, rs-FC can demonstrate this flexibility, and bilateral arm training can alter functional connections between sensorimotor brain regions and restore hemispheric imbalances in stroke recovery.

Zhang et al. (2016) examined the functional connectivity (FC) alterations between the ipsilesional primary motor cortex (M1) and the entire brain of stroke patients compared to normal controls, as well as in stroke patients preceding and following traditional rehabilitation and motor imagery therapy [23]. Relative to the controls, the FC in the patient group was substantially increased between the ipsilesional M1 and other areas of the brain.

This may be a form of compensation when interhemispheric FC balance is lost due to the stroke. Following the treatment, the FC between the ipsilesional and contralesional M1 increased, yet the FC between the ipsilesional M1 and other regions decreased. Specifically, the decreased FC between the ipsilesional M1 and left paracentral lobules—areas involved in the motion control and the sensation of the limbs—hinted at ipsilateral hemispheric FC recovery. The study revealed positive correlations between the FC change and the motor function recovery of stroke patients with hemiplegia, indicating that the FC could be used as a biomarker of motor function recovery. Moreover, Li et al. (2016) used repetitive transcranial magnetic stimulation (rTMS) over the ipsilesional M1 to increase CST activity, which aided in the recovery of motor function in stroke patients [24]. The authors attributed this to enhanced FC between the bilateral M1 and the contralesional supplementary motor area (SMA), which serves as an anatomical basis for motor recovery once M1 output is disrupted. Increased FC between the ipsilesional M1 and bilateral thalamus, which is a critical component of the extrapyramidal system and strongly related to motor coordination, was also noted, and this may be considered a possible compensatory mechanism for recovery of motor function after stroke.

On the same basis, Lefebvre et al. (2017) showed that FC increased exclusively in the somatomotor network after combined dual transcranial direct current stimulation (dual-tDCS)/motor skill learning relative to the period before intervention, which altered the spontaneous activity of the somatomotor network [25]. Consistent with previous studies, they supported the possibility that stroke could cause a net deleterious inhibition from the normal hemisphere to the damaged side. However, they also suggested another mechanism, namely, that stroke patients rely more heavily on their nondamaged upper limb in daily activities, which could strengthen the cortical FC of this limb.

Furthermore, Li et al. (2017), in their study of FC in subcortical stroke patients using conventional Western medical treatment and acupuncture, found a statistically significant association between pretreatment FC values between bilateral M1 and the percent changes in neurological deficit scores [26]. Given that ischemic stroke damage causes anatomical and functional changes in perilesional and distal brain regions, communication and connectivity between the two hemispheres were also influenced. Intra-hemispheric FC in regions of interests in the right contralateral hemisphere showed a considerable rise in post-stroke patients. This finding can be interpreted based on the mechanism of compensation for decreased FC of brain regions in the contralateral hemisphere to maintain motor function.

On the other hand, Guo et al. (2021) studied the motor networks of stroke patients with motor deficits after treatment with rTMS. Motor recovery could be due to the different functional recovery and reorganization that both high-frequency rTMS and low-frequency rTMS induced in the motor network [27]. Of note, after high-frequency rTMS and low-frequency rTMS, alterations were observed in the ipsilesional and contralateral hemispheres, respectively. This could be explained by the different processes of the various rTMS modes. High-frequency rTMS could promote cortical excitability, whereas low-frequency rTMS might reduce the abnormally excessive inhibition of lesioned M1 and support regrowth of the damaged hemisphere. As a result, the compensation of unequal interhemispheric excitability and connections may be related to motor recovery after stroke. A similar outcome was found by Chen et al. (2022), who investigated how coupled inhibitory and facilitative rTMS treatment affects early reorganization of motor networks in cerebral infarction patients [28]. They discovered that rTMS can induce functional remodeling of cortical motor networks by altering the intensity of intra- and interhemispheric FC. In addition, they note that FC changes were associated with a regaining of motor function and may serve as a future focus of neurorehabilitation interventions for early cerebral stroke patients.

### 2.1.3. Changes in FC and Structural Connectivity

Chen et al. (2013) demonstrated a correlation between the upper extremities FMA (UE-FMA) score and the left and right M1 rs-FC, as well as the white matter integrity in

the transcallosal M1–M1 tract [29]. However, they did not find any discernible association between WM integrity of the transcallosal M1–M1 tract and the rs-interhemispheric M1 FC. According to their theories, which are consistent with previous findings, the ipsilesional M1 may no longer have the same inhibitory effect on the contralesional M1 as before, and/or the contralesional M1 may have an excessively imbalanced inhibitory effect on the ipsilesional M1. It is therefore plausible that patients who recovered more quickly from their motor impairments had greater interhemispheric M1 connectivity and a more normal inhibitory effect from ipsilesional to contralesional M1.

The investigation by Kalinosky et al. (2017) successfully combined structural and functional connectivity techniques. While rs-FC alone can identify abnormalities in stroke patients that are associated with function, structuro-functional connectivity (SFC) detects the maximum FC to a voxel over various distances and may provide insight into network-specific processes underlying pathology and recovery [30]. The cerebellum, midbrain, and thalamus—integrative brain regions that are crucial for motor function—were found to have decreased SFC. Intrinsic SFC in the cerebellum was substantially linked with hand motor performance, serving as a marker of sensorimotor function. Additionally, stroke patients were found to have decreased intrinsic SFC between regions connected by longer fiber pathways, especially in areas associated with integrative cortical network nodes, including the cerebellum and prefrontal cortex: the center of the default mode network. This supports the possibility that alterations in these structuro-functional networks reflect compensatory or maladaptive mechanisms of reorganization and that the stroke-affected sensorimotor and cerebellar networks shift towards creating shorter-distance connections to the default mode and prefrontal network nodes.

Lin et al. (2018) discovered that structural and functional features may have good predictive value for initial motor impairment [31]. FC between motor areas improved in the long-term phase, but mainly in the first three months, which was substantially associated with motor function throughout the same period, but not at 12 months post-stroke. Therefore, the integrity of CST was found to influence interhemispheric FC, and recovery developed mainly within the first three months after the stroke, never reaching the optimal levels.

Research by Lee et al. (2019) shows changes in structural and functional connectivity after stroke measured using diffusion tensor imaging (DTI) and rs-fMRI [32]. Patients with mild stroke experienced restructuring of brain networks in motor-related areas, whereas the severe stroke group showed no significant alterations in FC, possibly indicating significant structural damage in the affected brain regions. The severity of motor impairment is influenced by alterations in myelination and axonal density, which may be due to differences in structural connectivity between the two groups. During poststroke reorganization, the severe group cannot successfully recover motor function through remyelination processes. Like other research, spared projections may be able to control paretic muscles post-stroke through perilateral reconfiguration of the motor network, which would lead to an increase in FC between specific motor-related brain regions such as M1 and the SMA.

With respect to rs-fMRI as a prognostic tool, Lu et al. (2019) support that FC of neuronal networks in the brain fluctuates dynamically because of synchronization of intrinsic neuronal interactions [33]. Compared to the baseline, the interhemispheric FC between ROIs was decreased at day 7, then gradually increased from day 7 to 90 and returned to normal at day 90. Of note, from day 7 to day 90 post-stroke, FA in the bilateral (ipsilesional, contralesional) CST increased, NHISS scores decreased, and FMA and BI progressively increased. The early decline in interhemispheric FC after stroke may point to the disruption of ischemic stroke-damaged networks. The elimination of transient hemispheric diaschisis and the formation of new axons forming new connections and projections might be associated with the restoration of interhemispheric FC. Therefore, the above results suggest that the response of the contralesional hemisphere to ipsilesional brain activity changes when the two hemispheres cannot communicate effectively at rest. Proliferative reorganization

in the ipsilesional and contralesional CST and functional reorganization in the SMN may support and promote neurological functional recovery after internal capsule infarction.

Similarly, the study by Xia et al. (2021) involving participants with mild to moderate stroke showed longitudinal improvements in their interhemispheric FC [34]. Notably, interhemispheric FC recovery largely took place within 4 weeks of stroke, making this the ideal time window for patients to experience interhemispheric FC reorganization. Considering these, it was found that increased interhemispheric FC and recovery of corticospinal pathways were positively correlated during the first 4 weeks after stroke but not between weeks 4 and 12.

*2.2. Prediction of Sensory Outcome Using rs-fMRI*

Bannister et al. (2015) found increased interhemispheric FC of S1 at 6 months in stroke patients with a lower degree of touch impairment that was not present at 1 month and lower interhemispheric connectivity in stroke patients with impaired touch compared with healthy controls [35]. The stroke group also showed FC at 6 months that was absent in the visual occipital regions and frontoparietal attention regions at 1 month. These results suggest a pattern of disrupted FC between somatosensory and other cortical regions that eventually exhibited some signs of recovery. Moreover, the improvements in the TDT score over time and the changes in FC between the 1- and 6-month periods were all located in the contralesional hemisphere. These emphasize the importance of an intact contralesional hemispheric FC, particularly in the somatosensory cortex. Moreover, Goodin et al. (2018) examined the rs-FC in stroke patients with both right and left hemisphere lesions and discovered that the right lesion group displayed greater FC in comparison with the left hemisphere group based on the connection between the contralesional left primary somatosensory cortex (S1) seed to the left superior parietal and mid-occipital regions [36]. This may indicate that connections between the somatosensory and visual networks are disrupted when only the left hemisphere, and not the right, is damaged. In addition, disruption of interhemispheric FC from both hemispheres was found regardless of the lesion location. Consequently, injury to the somatosensory system in one hemisphere not only affects interhemispheric inhibition but also exerts a negative effect on recovery in general.

## 3. Discussion

The current comprehensive analysis of the latest literature within the field of the application of rs-fMRI in stroke patients suggests that rs-fMRI can be a promising biomarker for the prediction of motor and sensory outcomes.

While task-fMRI focuses on mapping brain activity during specific tasks, rs-fMRI explores intrinsic connectivity patterns in the absence of explicit tasks. Both approaches contribute valuable information to the understanding of brain function and are often used together to provide a more comprehensive view of brain function. The combination allows for the identification of task-specific activations and the exploration of the underlying intrinsic connectivity patterns. Functional networks identified using rs-fMRI strongly overlap with the brain regions activated by task-fMRI [37], and spontaneous activity at rest correlates with trial-to-trail fluctuations in task-evoked responses [38] and behavior [39]. In contrast to task fMRI, resting-state fMRI is a non-invasive imaging technique, allowing for the examination of functional connectivity without the need of any external stimuli or tasks. The fact that the participants can simply rest with their eyes closed makes it suitable for a wide range of populations, including stroke patients. In addition, compared to task-fMRI, where participants perform specific cognitive tasks during MRI scanning, resting-state fMRI is relatively simple to acquire. The non-invasive character, the absence of specific experiment and participants' engagement during scanning, and the ease of acquisition facilitates the integration of resting-state fMRI into experimental protocols and clinical settings. For instance, a growing literature in brain tumor patients demonstrates that the results obtained from resting-state fMRI during pre-surgical language mapping can be

comparable to and, in some respects, superior to task-fMRI [40,41]. Thus, resting-state fMRI offers a great opportunity to study and understand the functional network changes caused by stroke, better assisting in patients' stratification in clinical trials and enabling rehabilitation interventions to be tailored to improve recovery.

Lesion volume or lesion location, age, stroke severity, and comorbid factors may influence stroke outcomes [42–48], but even mild stroke severity is not enough to provide an accurate prediction of favorable outcomes [49]. Thus, novel prognostic biomarkers are crucial for the development of personalized recovery treatment strategies [50] and the amelioration of patients' and caregivers' quality of life [51]. Brain plasticity remains a crucial issue in stroke rehabilitation, and advances in MRI techniques have the potential to improve the clinical and research opportunities and knowledge in that field. Clinical recovery, whether motor or sensory, is supported through reorganization of surviving brain tissue, alterations in interhemispheric lateralization, involvement of associative cortices linked to injured areas, and organization of cortical representation maps [52]. The variability of motor and sensory impairment after stroke has been associated with both structural [53] and functional alterations [52] in networks proximal and/or distal to the brain lesions. In contrast to task-related fMRI, rs-fMRI minimizes the subject's involvement and the interindividual differences in task performance, thus enabling the integration of rs-fMRI into clinical MRI and the monitoring of functional alterations during different post-stroke phases (e.g., subacute vs. chronic) and therapeutic approaches (e.g., physiotherapy, motor imagery, neuromodulation, etc.).

Some critical comments regarding methodological aspects of the available literature are necessary before reaching a conclusion. We found that some studies focused on motor outcomes using classic and more sophisticated approaches for the rs-fMRI analysis (Supplementary Material: Supplementary Table S2, section A). Other studies used rs-fMRI and focused on motor outcomes following specific rehabilitation interventions (Supplementary Material: Supplementary Table S2, section B), while other studies examined motor outcome in association with functional connectivity and structural connectivity (Supplementary Material: Supplementary Table S2, section C). However, few studies focused on sensory outcomes (Supplementary Material: Supplementary Table S2, section D). Most studies included patients with either ischemic stroke or hemorrhagic stroke. The total number of stroke patients included in all the studies varied significantly, ranging from 10 to 67 patients. Of note, the stroke patients were contrasted to demographically matched healthy individuals in most but not all studies, and none of the studies included a disease control group other than stroke patients. We also identified significant variability in the time of MRI acquisition post-stroke, ranging from a few days (i.e., 2–3 days) post-stroke to 60.0 +/− 69.6 months post-stroke. Most studies used 3T, while the rs-fMRI acquisition time (when available) ranged from 5 min to 10.25 min. Several approaches were applied to analyze the raw data, including independent component analysis (ICA), seed-to-whole brain FC analysis, region-of-interest (ROI) to ROI FC analysis, regional homogeneity (ReHO) analysis, amplitude of low-frequency fluctuations (ALFF) analysis, intrinsic structural–functional connectivity (iSFC) analysis, and static and dynamic functional network connectivity analyses. Some studies also included other MRI modalities and reported lesion analyses (location and extent), white matter hyperintensity analyses, structural integrity of white matter using diffusion tensor imaging (DTI), or cortical thickness of rs-fMRI ROIs. The most commonly used clinical measures were the National Institutes of Health Stroke Scale (NIHSS), the Fugl–Meyer Assessment (FMA), and the Barthel Index (BI).

Despite the available research on the application of rs-fMRI in post-stroke motor and sensory outcome so far, several questions remain unanswered, leaving room for further investigations. The studies included in the current comprehensive review are characterized by a high heterogeneity in terms of clinical characteristics. Thus, an attempt to organize this review based on clinical variables, including the type of stroke or lesion classification, might ultimately be less informative for the readers. A more in-depth

analysis of existing literature through a meta-analysis may further address this issue, and if possible, integrate information about lesion location and its impact at the connectome level (i.e., patterns of functional connectivity). The sample size of most studies was small, and together with the sample heterogeneity (e.g., stroke severity, post-stroke phase), the generalization of the current findings may be limited. Of note, different post-stroke phases are associated with different static and dynamic functional and structural features; thus, the inclusion of patients under different post-stroke phases may limit the clinical significance of these findings. In addition, the interventional studies included interventions of different durations, while the follow-up period differed among the studies. As a result, a dose–effect of interventions on FC patterns could not be easily found, whereas the durability of this effect requires further investigation in longitudinal studies with longer follow-up periods. Furthermore, even though the majority of the studies used the FMA for the evaluation of motor functional disability, gold-standard clinical measures (e.g., NIHSS) were not uniformly used in all the studies. From a methodological point of view, detailed descriptions of rs-fMRI protocols were not available for all the studies (e.g., duration of rs-fMRI sequence); thus, replication of the findings may not be easily obtained for all the studies. Finally, different and more sophisticated methodological approaches to rs-fMRI raw data may definitely yield informative details regarding the usefulness of rs-fMRI in research settings, yet the application in every-day clinical practice and prediction of single-subject prognosis has yet to be established. Thus, longitudinal studies with larger sample sizes and well-characterized groups in terms of clinical features are warranted to enable the identification of possible rs-fMRI biomarkers for the prognosis of stroke motor and sensory outcomes not only at group level but also—and more importantly—at the single-case level. In line with existing literature in pre-surgical planning [40,41], further studies contrasting the usefulness, sensitivity, and specificity of task-fMRI and resting-state fMRI in predicting stroke outcomes are also warranted.

## 4. Conclusions

With the advancement of MRI acquisition and data analysis methods, rs-fMRI emerges as a promising tool to study motor and sensory outcome in stroke patients and to evaluate the effects of different interventions, including recently used rTMS and tDCS. The implementation of rs-fMRI in stroke may further establish the predictive value of rs-fMRI in stroke clinical practice and provide valuable information for more accurate patient stratification into clinical trials and evidence-based clinical decision making.

**Supplementary Materials:** The following supporting information can be downloaded at: https://www.mdpi.com/article/10.3390/neurolint16010012/s1, Table S1: Search strategy and selection criteria for the most representative rs-fMRI studies in association to motor and sensory outcome after stroke which were included in the comprehensive review; Figure S1: Study flow chart of the process of selection the most representative rs-fMRI studies in association to motor and sensory outcome after stroke; Table S2: Representative rs-fMRI studies on motor and sensory outcome after stroke.

**Author Contributions:** Conceptualization, F.C. and C.K.; methodology, I.O.; formal analysis, D.T. and A.S.; writing—original draft preparation, E.M., E.K. and E.A.P.; writing—review and editing, A.T., F.C. and N.A.; supervision, K.V. All authors have read and agreed to the published version of the manuscript.

**Funding:** This research was funded by the project "Study of the interrelationships between neuroimaging, neurophysiological and biomechanical biomarkers in stroke rehabilitation (NEURO-BIO-MECH in stroke rehab)" (MIS 5047286), which was implemented under the action "Support for Regional Excellence" funded by the operational program "Competitiveness, Entrepreneurship and Innovation" (NSRFm2014-2020) and co-financed by Greece and the European Union (the European Regional Development Fund).

**Institutional Review Board Statement:** Not applicable.

**Informed Consent Statement:** The MRI (rs-fMRI and DTI) data (Figure 1) were acquired from a healthy control subject who provided written informed consent for the acquisition, analysis, and presentation of the data.

**Data Availability Statement:** No new data were created or analyzed in this study. Data sharing is not applicable to this article.

**Acknowledgments:** We thank the healthy control subject for participating in the acquisition of the MRI data. We also thank Advantis Medical Imaging for providing access to the Brainance MD platform to analyze the rs-fMRI and DTI data.

**Conflicts of Interest:** The authors declare no conflicts of interest.

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
