# Peer review of "A Comprehensive Review on the Role of Resting-State Functional Magnetic Resonance Imaging in Predicting Post-Stroke Motor and Sensory Outcomes"

_2035-8377, doi:10.3390/neurolint16010012_

Round 1

Reviewer 1 Report

Comments and Suggestions for Authors

The manuscript describes the utilization of fMRI for predicting post-stroke outcome.

The main topics of this paper will be important for the stroke rehabilitation.

Since it is a comprehensive review, it will be a summary of current knowledge, but I believe that the manuscript will be better if it is summarized in a way that is clinically useful. For example, classification by type of stroke or lesion classification.

Another point is that fMRI is often dependent on cognitive dysfunction and language impairment, so there are serious doubts as to whether it can provide enough information to be used as a prognostic factor for strokes in general.

Author Response

Dear Reviewer,

Many thanks for the time spent reviewing our manuscript.

Reviewer: The manuscript describes the utilization of fMRI for predicting post-stroke outcome. The main topics of this paper will be important for the stroke rehabilitation. Since it is a comprehensive review, it will be a summary of current knowledge, but I believe that the manuscript will be better if it is summarized in a way that is clinically useful. For example, classification by type of stroke or lesion classification.

Response: We thank the reviewer for dedicating time to review this manuscript and the encouraging comments with regards to the review and the importance of the topic for the stroke rehabilitation. We do acknowledge that type of stroke or lesion classification would be another way to organize the manuscript and provide clinically useful information to clinicians/researchers. However, the representative studies included in the current comprehensive review are characterized by high heterogeneity in terms of clinical characteristics, including type of stroke and lesion classification. Thus, we think that an attempt to organize this comprehensive review in terms of type of stroke or lesion classification might finally be less informative for the readers. In addition, our aim was to examine and present the utility of resting-state fMRI in predicting motor and sensory outcome, thus these were the sections originally selected to better address the main aim of the study. We do believe that a more in-depth analysis of existing literature by meta-analysis may further address this issue and if possible, integrate information about lesion location and its impact at the connectome level (i.e. patterns of functional connectivity). We now include the following sentences in the Discussion section and specifically the Limitations paragraph, “The studies included in the current comprehensive review are characterized by high heterogeneity in terms of clinical characteristics. Thus, an attempt to organize the review based on clinical variables, including type of stroke or lesion classification, might finally be less informative for the readers. A more in-depth analysis of existing literature by meta-analysis may further address this issue and if possible, integrate information about lesion location and its impact at the connectome level (i.e. patterns of functional connectivity).” Thank you.

Reviewer: Another point is that fMRI is often dependent on cognitive dysfunction and language impairment, so there are serious doubts as to whether it can provide enough information to be used as a prognostic factor for strokes in general.

Response: We thank the reviewer for giving us the opportunity to further elaborate on resting-state fMRI and its clinical applications and advantages over task-fMRI. We do apologize if this was not clear in the original version of the manuscript, since resting-state fMRI is independent of patients’ motor and/or sensory impairment, cognitive dysfunction, or language impairment. While task-fMRI focuses on mapping brain activity during specific tasks, resting-state fMRI explores intrinsic connectivity patterns in the absence of explicit tasks. Both approaches contribute valuable information to the understanding of brain function and are often used together to provide a more comprehensive view on brain function. The combination allows for the identification of task-specific activations and the exploration of the underlying intrinsic connectivity patterns. Functional networks identified by resting-state fMRI strongly overlap with the brain regions activated by task-fMRI (Fox et al., 2007 PMID 17704812) and spontaneous activity at rest correlates with trial-to-trail fluctuations in task-evoked responses (Fox et al., 2007 PMID 17920023) and behavior (Fox et al., 2006 PMID 16341210). For instance, a growing literature demonstrates that the results obtained from resting-state fMRI during pre-surgical language mapping can be comparable to and, in some respects, superior to task-fMRI (Lemee et al., 2019 PMID: 31568681; Ki Yun Park et al., 2020 PMID: 32735611). In contrast to task fMRI, resting-state fMRI is a non-invasive imaging technique, allowing the examination of the functional connectivity without the need of any external stimuli or task. The fact that the patients can simply rest with their eyes closed makes it suitable for a wide range of populations, including stroke patients. Compared to task-fMRI, where participants perform specific cognitive tasks during MRI scanning, resting-state fMRI is relatively simple to acquire. The non-invasive character, the absence of specific experiment and participants’ engagement during the scanning and the ease of acquisition facilitates the integration of resting-state fMRI into experimental protocols and clinical settings. Thus, resting-state fMRI offers a great opportunity to study and understand the functional network changes caused by stroke, assisting better patients’ stratification in clinical trials and enabling rehabilitation interventions to be tailored to improve recovery. In line with existing literature in pre-surgical planning (e.g. Lemee et al., 2019 PMID: 31568681; Ki Yun Park et al., 2020 PMID: 32735611), further studies contrasting the usefulness, sensitivity and specificity of task-fMRI and resting-state fMRI in predicting stroke outcome are also warranted.

The following sentences have been added in the Discussion section, “While task-fMRI focuses on mapping brain activity during specific tasks, rs-fMRI explores intrinsic connectivity patterns in the absence of explicit tasks. Both approaches contribute valuable information to the understanding of brain function and are often used together to provide a more comprehensive view on brain function. The combination allows for the identification of task-specific activations and the exploration of the underlying intrinsic connectivity patterns. Functional networks identified by rs-fMRI strongly overlap with the brain regions activated by task-fMRI [37] and spontaneous activity at rest correlates with trial-to-trail fluctuations in task-evoked responses [38] and behavior [39]. In contrast to task fMRI, resting-state fMRI is a non-invasive imaging technique, allowing the examination of the functional connectivity without the need of any external stimuli or task. The fact that the participants can simply rest with their eyes closed makes it suitable for a wide range of populations, including stroke patients. In addition, compared to task-fMRI, where participants perform specific cognitive tasks during MRI scanning, resting-state fMRI is relatively simple to acquire. The non-invasive character, the absence of specific experiment and participants’ engagement during the scanning and the ease of acquisition facilitates the integration of resting-state fMRI into experimental protocols and clinical settings. For instance, a growing literature in brain tumor patients demonstrates that the results obtained from resting-state fMRI during pre-surgical language mapping can be comparable to and, in some respects, superior to task-fMRI [40,41]. Thus, resting-state fMRI offers a great opportunity to study and understand the functional network changes caused by stroke, assisting better patients’ stratification in clinical trials and enabling rehabilitation interventions to be tailored to improve recovery.”

In addition, the following sentence is now included in the Discussion as a future direction,In line with existing literature in pre-surgical planning [40,41], further studies contrasting the usefulness, sensitivity and specificity of task-fMRI and resting-state fMRI in predicting stroke outcome are also warranted.”

Looking forward to your follow up comments.

Yours Sincerely,

Dr Tsiptsios

Reviewer 2 Report

Comments and Suggestions for Authors

Recovery after a stroke is the essence or critical element determining effective rehabilitation. Therefore, the article's topic should be considered important from a clinical point of view. In the introduction, the authors define these problems accurately and interestingly. 

The selection of a modern fMRI method is correct and in step with the trends of contemporary neurology; this fMRI examination can, therefore, provide therapeutically helpful information.

In general, the article is well-written and interesting. It covers the most critical aspects announced in the title "Role of Resting State Functional Magnetic Resonance Imaging in Predicting Post-Stroke Motor and Sensory Outcome". The only comment I would like to make is the need for a better-described methodology, especially the rules for including publications in the review. The authors mention the method too vaguely (p.33; 93-95); however, they provide a lot of data in the supplement. Nevertheless, the authors should present a search method for literature and its qualification for inclusion in the work. For methodological correctness, it would be good to include 1-2 sentences explaining, for example, the number of articles that meet the "keywords" conditions (what words?) were qualified in the work.

Author Response

Dear Reviewer,

Many thanks for the time spent reviewing our manuscript.

Reviewer: Recovery after a stroke is the essence or critical element determining effective rehabilitation. Therefore, the article's topic should be considered important from a clinical point of view. In the introduction, the authors define these problems accurately and interestingly. The selection of a modern fMRI method is correct and in step with the trends of contemporary neurology; this fMRI examination can, therefore, provide therapeutically helpful information.

Response: We thank the reviewer for dedicating time to review this manuscript and the encouraging comments with regards to the review, the utility of resting fMRI and the clinical significance of our findings as for the determination of effective rehabilitation in stroke patients.

Reviewer: In general, the article is well-written and interesting. It covers the most critical aspects announced in the title "Role of Resting State Functional Magnetic Resonance Imaging in Predicting Post-Stroke Motor and Sensory Outcome". The only comment I would like to make is the need for a better-described methodology, especially the rules for including publications in the review. The authors mention the method too vaguely (p.33; 93-95); however, they provide a lot of data in the supplement. Nevertheless, the authors should present a search method for literature and its qualification for inclusion in the work. For methodological correctness, it would be good to include 1-2 sentences explaining, for example, the number of articles that meet the "keywords" conditions (what words?) were qualified in the work.

Response: We thank the reviewer for giving us the opportunity to address this methodological issue. Based on manuscript’s type instructions provided by the Journal for Comprehensive reviews and considering that our study is not a systematic review, we did not extensively describe methodological aspects but all studies included in this comprehensive review are presented in detail in the supplement for further reading. According to the reviewer’s suggestion, we included the search method for literature and further information in the supplement for methodological correctness. The following highlighted phrase is now included in the main text , “To address the main aim of this comprehensive review, we reviewed most representative human studies (cross-sectional or longitudinal) during the last decade, which used rs-fMRI and focused on stroke motor and sensory outcome. (Supplementary Material: Supplementary Table 1, Supplementary Figure 1, Supplementary Table 2). In addition, the following supplementary Table and Figure is now included in the Supplement. Thank you.

Supplementary Table 1. Search strategy and selection criteria for the most representative rs-fMRI studies in association to motor and sensory outcome after stroke which were included in the comprehensive review

Search strategy:

A literature research of two databases (MEDLINE and Science Direct) was conducted by one investigator in order to trace all relevant studies published between January 1, 2012 and December 31, 2022, using either “resting state functional magnetic resonance imaging” as keyword or related term “resting state fMRI ”as a search criterion.  Also, the terms “stroke prognosis” or “stroke outcome” or “stroke recovery” were used as a second search criterion. The retrieved articles were also hand searched for any further potential eligible articles. Any disagreement regarding screening, or selection process, was solved by a second investigator until a consensus was reached. Supplementary Figure 1 presents the review flowchart.

Selection criteria:

Only full-text original articles published in the English language that focused on motor and/or sensory outcomes in stroke patients were included for further analyses. Secondary analyses, reviews, case reports, guidelines, meeting summaries, comments, unpublished abstracts, or studies conducted in animals were excluded. There was no restriction on study design or sample characteristics.

Data extraction:

Data extraction was performed using a predefined data form created in Excel. We recorded author, year of publication, number and age of participants, study design with regards to rs-fMRI acquisition (cross-sectional, longitudinal), type of stroke, time of rs-fMRI acquisition, main rs-fMRI parameters (MR field strength, duration of rs-fMRI acquisition), type of rs-fMRI analysis (ICA, seed-to-voxel, ROI-to-ROI analysis, etc), functional networks examined (motor, sensory, both), clinical scales utilized and main findings in terms of motor and sensory outcome and rs-fMRI networks.

Data analysis:

No statistical analysis or meta-analysis was performed.

Supplementary Figure 1. Study flow chart of the process of selection the most representative rs-fMRI studies in association to motor and sensory outcome after stroke.

Looking forward to your follow up comments.

Yours Sincerely,

Dr Tsiptsios

Round 2

Reviewer 1 Report

Comments and Suggestions for Authors

Thank you for early correction.

I think the content of the discussion has become deeper and easier to understand.